# An IoT-Based Anonymous Function for Security and Privacy in Healthcare Sensor Networks

**DOI:** 10.3390/s19143146

**Published:** 2019-07-17

**Authors:** Xiao Chun Yin, Zeng Guang Liu, Bruce Ndibanje, Lewis Nkenyereye, S. M. Riazul Islam

**Affiliations:** 1Facility Horticulture Laboratory of Universities in Shandong, Weifang University of Science & Technology, Shouguang 262700, China; 2College of Computer Science and Engineering, Shandong University of Science and Technology, Qingdao 266590, China; 3Research and Development Center, Cyber Threat Intelligence Lab, YangJae Innovation Hub, 114 Taebong-Ro, Seocho-Gu, Seoul 06754-601, Korea; 4Department of Computer and Information Security, Sejong University, Seoul 05006, Korea; 5Department of Computer Science and Engineering, Sejong University, Seoul 05006, Korea

**Keywords:** IoT, security, privacy, anonymous function, healthcare, wireless sensor networks

## Abstract

In the age of the Internet of Things, connected devices are changing the delivery system in the healthcare communication environment. With the integration of IoT in healthcare, there is a huge potential for improvement of the quality, safety, and efficiency of health care in addition to promising technological, economical, and social prospects. Nevertheless, this integration comes with security risks such as data breach that might be caused by credential-stealing malware. In addition, the patient valuable data can be disclosed when the perspective devices are compromised since they are connected to the internet. Hence, security has become an essential part of today’s computing world regarding the ubiquitous nature of the IoT entities in general and IoT-based healthcare in particular. In this paper, research on the algorithm for anonymizing sensitive information about health data set exchanged in the IoT environment using a wireless communication system has been presented. To preserve the security and privacy, during the data session from the users interacting online, the algorithm defines records that cannot be revealed by providing protection to user’s privacy. Moreover, the proposed algorithm includes a secure encryption process that enables health data anonymity. Furthermore, we have provided an analysis using mathematical functions to valid the algorithm’s anonymity function. The results show that the anonymization algorithm guarantees safety features for the considered IoT system applied in context of the healthcare communication systems.

## 1. Introduction

Nowadays, medical caregivers are able to monitor the patient’s status in real-time and the relevant status can be updated time-to-time using applications and infrastructures. The connectivity protocol based on the IPv6 low-power wireless personal area network is the most used in the IoT environment to support healthcare mobility via wireless approaches [1]. Modern healthcare is reshaping the presence and evolution of the IoT that support technology, economy, and social networks. The IoT state-of-the-art reflects an inter-connection of people, anything, accessing any service anytime, anywhere and on any network. It is seen as a megatrend technology in information and communication technology (ICT) that is influencing the entire business spectrum with more advantages going beyond machine-to-machine (M2M) states [2]. The solutions provided by the IoT are now exploitable in multiple areas of applications like logistics, industrial control, smart cities, transportation, retails and healthcare systems [3,4,5,6,7].

Among the aforementioned applications areas and others, the attention is given to the healthcare system, which represents the most attractive application for developers and consumers [8]. The main reason is that the human being is much involved for applications such as elderly care, fitness programs, remote health care monitoring and chronic diseases surveillance. Figure 1 gives a generic illustration of a body area network consisting of IoT medical sensors.

The health devices collect and transmit patient health data to the medical service providers for data analytics and visualizations to facilitate health monitoring and treatment. As shown in Figure 1, the sensors can be embedded into the body. They are smart electronic devices equipped with a micro-controller to compute different functions. Therefore, in IoT-based health care, the devices are inter-connected, embedded with software and use a wireless communication system to exchange the data [9].

However, security and privacy are highly discussed in such context given that devices or/and software’s compromises directly leads to the safety of the user (here, patient) and can thus cause harmful consequences, even a death. Most of the privacy and security solutions for healthcare systems are discussed in Section 2. Using wireless technologies, any user with his sensitive data such as bank transactions, health data, and email should exchange via a platform that provide the user’s privacy and ensures the security of their information. Nevertheless, in a centralized system such as IoT-based health care, the services providers can access the data and have capabilities to possibly perform a-priori or a-posteriori control or reveal the sender’s message in their system to other entities, irrespective of the concerned privacy level. Unfortunately, such a situation usually happens when the relevant institutions abuse the user information. With this comportment, it is obvious that privacy and data protection need more attention as long as any misusage can lead to a threat. Therefore, security experts have developed applied solutions to support the confidentiality of the information that protects sensitive information in IoT-based health-care domain. With the said issues, this paper presents an approach to the solution based on the anonymization process that provides features such as privacy and security of sensitive health dataset.

On that, the main contributions of this paper are the following: We develop an IoT algorithm that provides an anonymous function.We present a strong mathematical basis to prove the privacy and security functions that protect the data being exchanged over the internet using a wireless communication system. This method follows the homomorphism equation via the Identity-based Encryption (IBE).We provide an algorithm on computational complexity to evaluate the proposed anonymization algorithm whether it satisfies the complexity requirements during algorithm execution.Conversely, the proposed method has a couple of limitations that introduces some opportunities for further research in the IoT-based health care system:The anonymization algorithms work within a standalone healthcare system and third party. As many services and providers gradually adopting cloud-based operations, further research are required to overcome the limitation in our algorithm. This would require an additional function to communicate with a cloud provider with anonymization as security service.Taking of privacy for data anonymization into account, the user should have the ability to choose his anonymous parameters. However, our method does not offer the option that is reserved for future work.

The remainder of this paper is organized as follows: related work is briefly summarized in Section 2, while the proposed algorithm is presented in Section 3. The proposed IBE related to our algorithm is outlined in Section 4, along with the mathematical basis. The final section concludes our work.

## 2. Related Work

Since the Internet of Thing has been emerging in the health care system, personal health records have become prey for cyber-attackers or hackers. This is a dangerous situation because any data breach leads to exposure of sensitive information and patients can no longer trust the system nor the medical staff anymore. Consequently, the patients may take drastic measures such as a denial of any healthcare service, hiding information, or staying home to avoid seeking medical help [10]. In this section, we present different solutions that are applied in IoT-based health care to solve the issue of privacy and security of patient records.

Lightweight solutions (to overcome resources constrain in IoT devices) that support authentication and authorization have been proposed by Lee et al. in [11], where they develop a method to encrypt the data using logic operations for the encryption processes. Gong et al. [12] developed a scheme that includes a homomorphism system enhanced from the DES algorithm with a model system related to the lightweight scheme. In the same research way, a protocol for IoT in the electronic health is proposed by Seyed et al. [13] and outlined security features like authentication, key agreement, access control, and energy-efficient are available.

Data anonymizing with denaturing framework has been developed with the following aspects: (a) the users have possibility to define rules before the algorithm is deployed, (b) personal data masking system, (c) analytic system to allow denaturing, deletion inference anonymization and mobility data privacy function and a wide range of research has been proposed to satisfy these features [14,15,16,17,18,19].

Furthermore, interesting research by Langheinrich [20,21] has contributed to the field of privacy-preserving security. The work consists of a system in which a customer or user has some options to select instead of having negotiations with a computerized procedure in order to have an adequate agreement. The architecture provides data privacy and ensures that collected data is kept confidential by notifying the user what kind of data has been collected. With this acknowledgment, the user has the ability to decide on the actions to be taken regarding the data. The same author [21] incorporated a function to preserve the privacy ubiquitously. The architecture is mainly composed by four elements: (a) the choice and consent provided by machine-readable privacy policies, (b) a notice mechanism based on a policy announcement, (c) access control supported by the privacy proxies, (d) resource protection provided by a policy-based on data access.

Kavenesh et al. [22] proposed a framework that models and considers the main privacy concepts suitable for the healthcare applications in IoT. The proposed compliance scale presents essential privacy principles that can be considered in the development of novel IoT health applications. The proposed compliance scale would be significant for policymakers and applications developers to measure understand and respect the privacy principles of consumers towards novel IoT-based health applications.

A Privacy Protector framework that protects collected data from the patient has been developed in the IoT network. This framework consists of sensors that collect the patient’s body data, a communication service provider to prepare security scheme, a storage system to receive data from sensors and finally a system of data access control to get access to the user data. The main idea is based on secret sharing and shares paring for patients’ data privacy [23]. Besides anonymization techniques, other methods to protect medical data have been presented in previous researches. A Context-Aware Access Control (CAAC) models have been developed, extending the basic Role-Based Access Control (RBAC) model where the author develop methods based on the access and privacy control policies to manage sensitive data and determine whether users’ requests to limit data access permissions based on the contextual conditions as developed in the recent works [24,25,26]. Furthermore, Kayes et al. [27,28] have developed CAAC models including features such as sensitive and streaming data management which are applied in today’s IoT-based smart spaces. In their works, they considered a wide variety of contextual conditions, for example, the situational and relationship context, utilizing the process of inferring implicit knowledge from the currently available context information.

## 3. Proposed IoT-Based Anonymization Algorithm for Security and Privacy in Health Care

### 3.1. System Model and Overall Description

This section describes in detail the proposed anonymization algorithm that preserves security and privacy in IoT-based health care system. Two algorithms compose the whole system and the main steps are depicted in Algorithm 1. The description of some parameters are given in Table 1 and other parameters are described throughout the algorithm.

First of all, we describe the system model which includes two main parts: the Healthcare System (HYsy) and HealthThird Party (HTP) as given in Figure 2. The HSys includes the data owner such as patients and physicians with their databases (DB). Furthermore, the system possesses a security engine to encrypt the data with the defined parameter. The HTP host the anonymization engine with the parameters to perform the data anonymization process. The anonymous data is available once all steps described in the algorithm are executed. In the end, the HTP can return the anonymized data to the HSys where the corresponding user can then decide when and where to share his data.

Algorithm 1 is the overall scheme from the step in which the users interact via the HSys using their sensor or mobile devices to exchange the HDs. Let HSys be the Health care System environment (e.g., clinic, hospital, remote care …) where the data owners are playing a part in. Further, let be Sick Person <Sp> and Physician <Ph> be the users in the HSys. The system requires each user participating to have a secret key pair <m_s_, n_s_>.

**Algorithm 1:** Overall Algorithm (Tripartite: <User, HSys and HTP>)

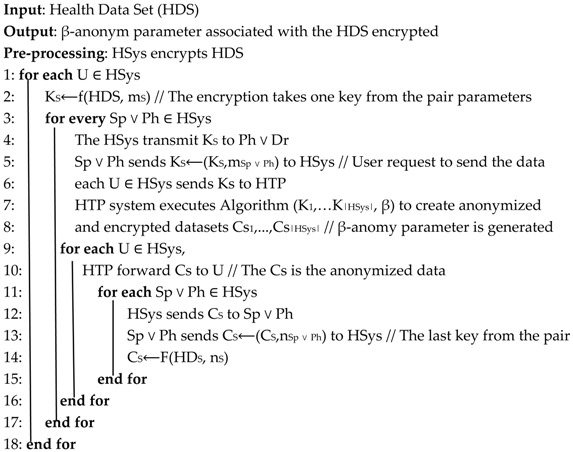



**Statement** **1.**
*Each user $U\;\in \;HSys\;\overset{Possesses}{\leftarrow }\;$ a private key pair $\langle m_{s},n_{s}\rangle$ and private dataset denoted as $HDS_{s}$ with s generated by HSys.*


Statement 1 denotes that a user that belongs to the healthcare system has a private key pair to be used for cryptographic functions to secure his data. Within his private keys, the system generates its random private key too.

During the first operation of the session when the user wants to exchange the data, it is encrypted by the HSys’ key. The anonymization comes in when the data is sent out of the HSys to the HTP or collaborating hospital or other health organizations. In this case, the algorithm generates an anonymization parameter β that is assigned to the encrypted data. The parameter is required for additional steps and the HTP returns a response message to the HSys that the data has been anonymized with β. The user with secret pair key is the only one to whom the data can be disclosed as long as he holds the secret pair keys. At this end of operations, the user system holds his encrypted (Cs) data with a β-anonym parameter.

### 3.2. Process of Health Data-Set Anonymization

As long as the data is being exchanged inside of the HSys, the users are confident. However, when IoT comes in, the data handling or exchange becomes problematic with all attack types over the internet. The anonymization process is triggered when the user is in the position of sending his data to the HTP or other collaborating organizations. The negotiation is tripartite: User, HSys and HTP and final decision is made by the HTP where the β-anonym parameter is generated to the other entities.
**Statement** **2.**Each user U ∈ HSys ⇄Receiveing a β−anonym parameterSending HDS to HTP using HSys, the data is encrypted and anonymized.

Statement 2 indicates that a user belongs to the healthcare systems and can send his data to the third party. Before that, the user makes sure that his data is encrypted and anonymized.

The HSys, after getting the data from the user (here: <Sp> or <Ph>), the system encrypts the data and afterward, transmit the cipher to the designated HTP. The system then analyzes the request and generates an anonymized dataset. In the beginning, the user systems pre-analyze which data is designed to be anonymized so that it is not revealed to the rest of the networks. The proposed method’s details are given in the pseudo-code of Algorithm 2.

### 3.3. Description of the Algorithms

As the very first step consists of a request of anonymization function, in this case, we consider the response to the HSys as health data set from the HTP. Therefore, to get back or construct the responses, heuristic and approximation methods have been utilized for data allocation.
To boost the allocations number with a no-null response, the heuristic method is designed and can be observed in Algorithm 2. From step 5, some *d-encrypted* samples are allocated to the HSys from its submission to its response. This operation is done until each considered health system (in a distributed environment such as IoT-based healthcare [29,30]). It is not allowed that in case a sample is put on the reply, a similar action cannot be computed on the rest of the responses in the health system. Partial data (samples) is allocated to the HSys, which performs the big data of the user through the probability of the prediction. The observation of a chosen sample has probably a low value when it is randomly submitted. The steps 16 to step 18, show such case when it is not possible for a response to be allocated *d* samples, therefore the systems ensures that null samples are allocated. During this process of the operation, the algorithm guarantees that any location should provide a user data within *d* size otherwise *d–anonymity* rules are not satisfied.The approximation method can be seen at step 21 where it processed using a minimally sized submission allocation to the rest if the encrypted sample from the HSys submission. According to a distributed IoT-based healthcare in [29,30] where more than one health system is interacting, these samples are removed from other entities in which all system have no samples to send until no more system can be allocated samples to release. When all steps of the algorithms are completed, the HTP broadcast a message to each contributing HSys that a sample has been designed to be public or/and which one is anonymized.

### 3.4. Algorithm Complexity Computation

The evaluation of the algorithm is done by the so-called time complexity in algorithm execution procedures. “Time complexity of an algorithm quantifies the amount of time taken by an algorithm to run as a function of the length of the input.” For instance, in step 1, the health datasets are reduced considering the result of the intersection tests. This process is computed in O(|HSys|logHSys|) assessments. For step 6, in every HSys the user data is assigned d answers but, they are no longer in the session because they are cleaned from all participating entities in the health network system. Consequently, this necessitates O(|HSys|) phases at the condition of d to be fairly enough the smallest value. The step 18 shows clearly that the datasets are zeroed due to the lacking of size, the complexity is O(|HSys|) of the linearity function. The complexity at step 21 is O(log|HSys|) after considering the assignment of the data during round two in addition to data cleaning. The complexity of the algorithm, once the maximum steps are computed, is O(log|HSys|). This result is within the allowed standard complexity during algorithm execution.

According to [31,32,33], the complexity analysis of an algorithm is determined by the resources like time and storage which are required to execute the algorithm. Furthermore, most algorithms are designed to be executed based on the inputs of random length or size. Often, the complexity is defined as a function of the input or size given a number of fundamental steps (in here: time complexity) with sometimes the fundamental storage (called space complexity). To this end, notations such “O” (Big O) and theta notation (Θ) are usually utilized and in this paper, we have used only big-O. For example, the logarithmic time {noted O(log(n))} in the binary search operation running, means that there a list is a proportional number of steps being searched off to the length logarithm.

**Algorithm 2:** Anonymization Process (α, Ω, and β-anonym parameter)

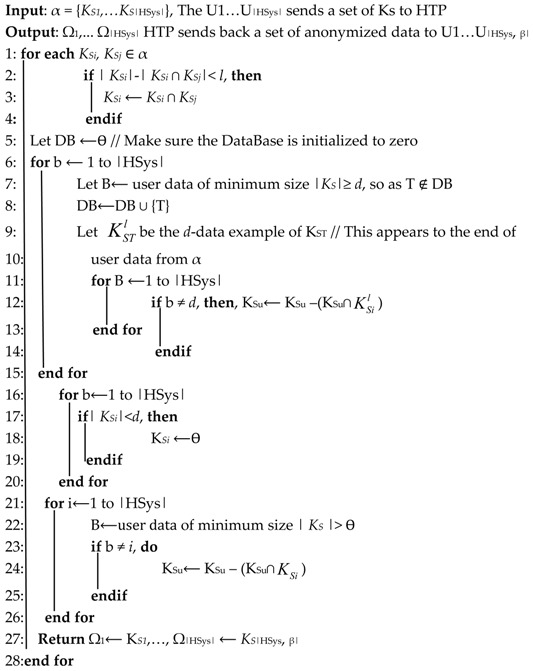



From this observation, we have clarified our results of the complexity analysis in the following algorithms: quicksort (step 1: O(n log n)), linear search (step 6: O(n)), Linear search (step 18: O(n)) and binary search (step 21: O(log n). Figure 3 gives the overview classification of the result from our algorithm where Y-Axis represents the operations in the algorithms and X-Axis represents the elements or inputs in the algorithm.

## 4. Algorithm Evaluation Based on Mathematical Concepts

This section describes an evaluation of the proposed algorithm using a mathematical approach based on an encryption process to prove the correctness in a communication environment. The evaluation follows Shamir idea in [34] where we incorporate the main parameters used in the IoT-based healthcare algorithm. The idea is based on the IBE algorithm as it supports anonymity functions for sensitive information passing through wireless communication.

### 4.1. Preliminary

The notion of the algorithm based on IBE is able to give the users the ability to utilize their identity to generate the public key to encrypt the data in addition of an easy approach of public key certificates management. The infrastructure managing public key is usually used in case of the non-cryptographic model. This infrastructure is used to legitimate the public key and it is called certificate authority where it authenticates and distributes to the users their matching certificate of the public key. During IBE key exchange session, a user can utilize any string to encrypt his data but there exist other encryption methods which do not require a Public Key Encryption (PKE) infrastructure.

### 4.2. Generation of Homomorphism Equation via IBE

Fundamentally [35], the IBE method is characteristically a tuple-algorithms denoted as <I-B-E> = (Set-up, Extract, Encr, Decr) and described as follows:Set-up: The responsible of key generation runs Setup to generate a secret parameter a where it receives an ensemble of parameters (parames) and main key. In this early stage, the parames comprise a space of message with limitations denoted P together with L as a crypto-message. To have the main key as a private element, the PKG is involved.Extract: This algorithm is about input parames, including the main key with Id belongs to [0, 1]* where it receives f as a private key. During this phase, the id and public key are an arbitrary sequence with f as a private key.Enc: Basically, any algorithm with the encryption phase takes some parameter to encrypt and in this case, they are as follow: The input is <parames, Id, and P_0_ ∈ P>, the output is a cipher-message <L_0_ ∈ L>.Dec: In the same way, the decryption is the counter-part algorithm with parameters such as the input is <parames, L_0_ ∈ L, the private key *d*>, where the output is <P_0_ ∈ M>.

The <I-B-E> is designed to be reliable which implies that any *Id*, during the extraction of *f* as private key (during the 2nd algorithm: extract), the operation is as follow: Dec (parames, L_0_, *f*) = P_0_, with P_0_ ∈ P and *L_0_* = Encry (parames, *Id*, P_0_).

#### Homomorphism Equation

The proposed anonymized algorithm in the IoT-bases healthcare needs a homomorphism equation (encryption) which is mathematically computed using the IBE. Fully homomorphic encryption can take two operations: addition and multiplication [36], in this paper, we define a homomorphic equation with two products:
Init step: Let be *ε* a prime order of <Ꝕ_1_, Ꝕ_2_ > the two cyclic groups, and $\overset{⌢}{u}$: Ꝕ_1 X_ Ꝕ_2_ → Ꝕ_2_ as an acceptable bilinear map of group Ꝕ_1_ with generator *Q*. Now, consider *s* a secret factor and *t* to be ϖ-bit prime. Assuming that *τ-bit* strings represented all identities (where *τ* is polynomial in ϖ). With Β: [0, 1]*^τ^* → Ꝕ_1_ as an algorithm for data mapping (B is the hash function). Next, ∀ δ ∈ {0, 1, …, ε −1}, the computation of the public key relies on a random choice of δ homogeneously and it gives *PbKey =*
*δQ*. From this computation, the only parameter δ is the secret main key but the rest of the parameters remain public.Secret Key Generation Step: This is the step where the *<I-B-E>* scheme computes the main keys such as secret (or private) and its corresponding public key as follow: For *∀ Id*, ∃ *Φ_Id_ = B* (*δ_Id_*, *Id*) and *Θ_Id_ =*
*B* (*δQ*, *Id*) as main secret and public keys respectively. Here B is a cryptographic hash function to compute the keys.Cipher Process Step: The cipher message is the result of the secret key with the encryption process over the message itself from the sender. This operation is done as follow: Let *σ* be a sample data *∈*
*Ꝕ_2_* and *Id* the user identity, *Ψ**=**E**nc_Id_* (*σ*, *μ*) *=* (*σ**⋅*
$\overset{⌢}{u}$(*μ_Id_*, *Id*), *ΘId*), where *μ*
*∈* {0, 1, …, *q* − 1} is a secret parameter which is arbitrary selected consistently. The cipher from this encryption process generates a result denoted *Ψ*.Decipher Process Step: To complete the *<I-B-E>* full cycle algorithm, the system must provide a function to retrieve the original message from the sender. This operation is called decryption. Let *Id* be the user identity and *Ψ* the cipher such as *Ψ =* {$\Psi^{\prime }$, $\Psi^{\prime \prime }$}, the decryption process is a function of ρ (plaintext) and *Ψ* (cipher):(1)$$\rho_{\Psi }=\frac{\Psi^{\prime }}{\overset{⌢}{u}(\Phi_{Id},\Psi \prime \prime )}$$

The Equation (1), leads to the homomorphism Equation (2), which is needed for the anonymization algorithm in the further steps. The homomorphism equation satisfies the following:(2)$$\begin{array}{ll} Enc_{Id} & =(\sigma^{\prime }⊗\sigma^{\prime \prime },\mu^{\prime }⊕\mu^{\prime \prime })\\ & =Enc_{Id}(\sigma^{\prime },\mu^{\prime })Enc_{Id}(\sigma^{\prime \prime }⊗\mu^{\prime \prime }) \end{array}$$

### 4.3. Theoretical Proof with Mathematical Analysis

The theoretical proof presents the mathematical concepts, which are applied in order to generate the anonymization function used in the IoT-based healthcare algorithm.

Therefore, we recall the main parameters involved in this process such as Users such as *Sp* and *Ph* with *Id_Sp_* and *Id_Ph_*, which are the ids of the users respectively in the HSys. We assume that there is no exploitation of the user‘s health data between the system communication and HTP as they may collaborate to expose the data.

Considering HSys, there are ***η*** users, the total of users that are playing part in the communication system with <N> the total number of all corresponding *Ids*. Given that a single identification is assigned to every user has a unique identity *a* (unique identity usually is such as ID-NUMBER), we define a network <V> with all identities and users such <a> user has only and only one identity <1> with the condition that two users in <V_(a,b)_> cannot have matching identity defined as follows:$$\begin{array}{l} \forall \left(a,b\right)\in \left\{Users\right\}\neg \exists V_{\left(a,b\right)}=V_{\left(1\right)}\\ V=\left\{Users_{\left(a,b,\ldots \eta \right)}\cup Ids_{\left(1,2,\ldots ,N\right)}\right\} \end{array}$$

While submitting the data, the system performs a comparison task. Let the user <a> in V (V_(a)_) compares his identity <1>, in *n*-th data transfer rounds that correspond to the initial process, the system performs the comparison, if the condition is satisfied, the <a> user computes and send:(3)$$\chi_{1,a}=Enc_{Id_{Sp}+Id_{Ph}}(\zeta_{a},V_{1,a})$$

This is the operation in Equation (3) where the system sends an encrypted χ dataset where $\zeta_{a}$ is a sample user health dataset taken from the HDS and $V_{1,a}$ is a pair of user and his identity with an arbitrary selection process from the health care system configuration. The condition V(a)≠1 is checked and if it is satisfied, the user system computes and submit the following:(4)$$\chi_{1,a}=Enc_{Id_{Sp}+Id_{Ph}}(1,V_{1,a})$$

Equation (4) is a particular case where we specify the user identity <1> which gives to the system a possibility to compute all submission rounds of his data set:(5)$$\chi_{1}=\prod_{a=1}^{N}\chi_{1,a}$$

Equation (5) represents a computation of all datasets until the last round submission. The HSys then forward all t $\chi_{1}$ to the HTP. The system in the third party will compute the following expression and send back it the HSys for further steps:(6)$${\hat{\zeta }}_{1}=\chi_{T_{Id_{Sp}}}(\chi_{1})$$

Equation (6) shows us that the system can identify the ID of the user who is, in this case, the patient and for now, the anonymization process is getting started. Moreover, it is remarkable that {T} value includes the expression to specify that this is not a replicated data as described in Algorithm 2.

Suppose that: $\chi_{1}=\left\{{\chi^{\prime }}_{1},{\chi^{\prime \prime }}_{1}\right\}$ in this assumption, the health care system in the IoT configuration will compute the following:(7)$$\tilde{\zeta }=\chi_{T_{Id_{Sp}}}(\hat{\zeta },\chi_{1}{}^{\prime \prime })$$

The result in Equation (7) shows that partial data $\tilde{\zeta }$ is encrypted with the owner data *Id*; in this case, it is <Sp>. This expression leads us on the following theorem where all involved users with their data are assigned a random number.
**Theorem** **1.**The correctness of the anonymization function in the algorithm is true; that is, assuming that all involved parties follow the rule, then $\left({\tilde{\zeta }}_{1},{\tilde{\zeta }}_{2},\ldots ,{\tilde{\zeta }}_{n}\right)$ is a permutation of $\left(\zeta_{1},\zeta_{2},\ldots ,\zeta_{n}\right)$
**Proof** The demonstration of the provability clarify it as follows:(8)$${\tilde{\zeta }}_{{}_{1}}=\zeta_{T_{Id_{Sp}}}(\hat{\zeta },\chi_{1}{}^{\prime \prime })=\frac{{\hat{\zeta }}_{1}}{\hat{u}(\Phi_{Id_{Sp}},\chi_{1}{}^{\prime \prime })}=\frac{\chi_{T_{Id_{Dr}}}\left(\chi_{1}\right)}{\hat{u}\left(T_{ID_{Sp}},\chi_{1}{}^{\prime \prime }\right)}=\frac{\chi_{1}}{\hat{u}\left(T_{Id_{Sp}},\chi_{1}{}^{\prime \prime }\right)\hat{u}\left(T_{Id_{Ph}},\chi_{1}{}^{\prime \prime }\right)}$$

In Equation (8), the system takes into consideration of all involved users (Sp and Ph) as we stated that the anonymization rule must be applied on each entity-playing role in the IoT system. 

The last computation is based on the condition that user <a> and its id <1> satisfies the condition such as Va=1 for a(1) as its value, this will transform the Equation (8) into the following global Equation (9) and final result:(9)$$\begin{array}{ll} {\hat{\zeta }}_{1} & =\frac{\chi_{1}}{\hat{u}(T_{Id_{Sp}},\chi_{1}{}^{\prime \prime })\hat{u}(T_{Id_{Ph}},\chi_{1}{}^{\prime \prime })}=\frac{\prod_{a=1}^{N}\chi_{a,1}}{\hat{u}\left(T_{Id_{Sp}},\chi_{1}{}^{\prime \prime }\right)\hat{u}\left(T_{Id_{Ph}},\chi_{1}{}^{\prime \prime }\right)}\\ & =\frac{Enc_{Id_{Sp}+Id_{Ph}}\left(\zeta_{a(1)},V_{1,a}\right)\prod_{a\neq a(1)Enc_{Id_{Sp}+Id_{Ph}}}\left(1,V_{1,a}\right)}{\hat{u}\left(T_{Id_{Sp}},\chi_{1}{}^{\prime \prime }\right)\hat{u}\left(T_{Id_{Ph}},\chi_{1}{}^{\prime \prime }\right)}=\zeta_{a(1)} \end{array}$$

With the permutation, operations of a(1) we deduct that the permutation function is applied on (1, 2… *N*), the result is then a permutation of the following expression: $\left(\zeta_{1},\zeta_{2},\ldots ,\zeta_{n}\right)$, which proves Theorem 1 of anonymization algorithm.

## 5. Conclusions

With the growth of the IoT-based healthcare system, extensive studies offering applicable solutions in the field have been developed and others are still going on. Considering such an environment, an immense volume of data is transmitted over the public network among the patients, physicians, nurses, and relevant health organizations. Therefore, it is highly important to assure the safety of the data owner to avoid an unwanted situation. This paper proposed the development of a theoretical approach that ensures the security and privacy of sensitive data for the considered IoT environment.

The proposed algorithm provided required security features such as privacy or confidentially for the user’s data that is transmitted within the health care network. When the user sends his information to be used by the third party via a given health network, the encryption process is firstly executed using a key from the key pair and the system request a response to the third party in which the anonymization function generates a value to anonymize the encrypted data set.

In the work, we showed that our proposed scheme guarantees the anonymity function where the algorithm computes the conditions and then executes the anonymization procedure on the healthcare data. In addition, we demonstrated that the algorithm satisfies the computational complexity requirements of the execution of all steps. Lastly, a proof based on a mathematical analysis has been developed to demonstrate that the proposed algorithm ensures the veracity and can be a real application to secure the IoT technologies for the health care network using wireless communications. For future work, we intend to implement the proposed approach in a practical environment (using healthcare sensors). The experiment results can, therefore, be used for evaluation and comparison with other existing methods.

## Figures and Tables

**Figure 1 sensors-19-03146-f001:**
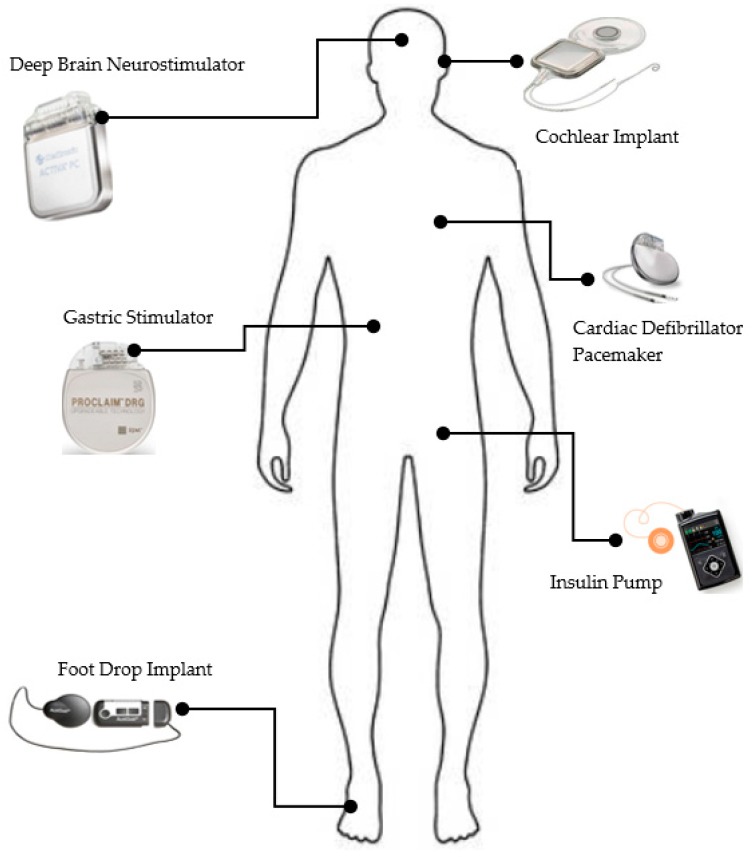
Overview of a patient’s body with IoT medical sensors.

**Figure 2 sensors-19-03146-f002:**
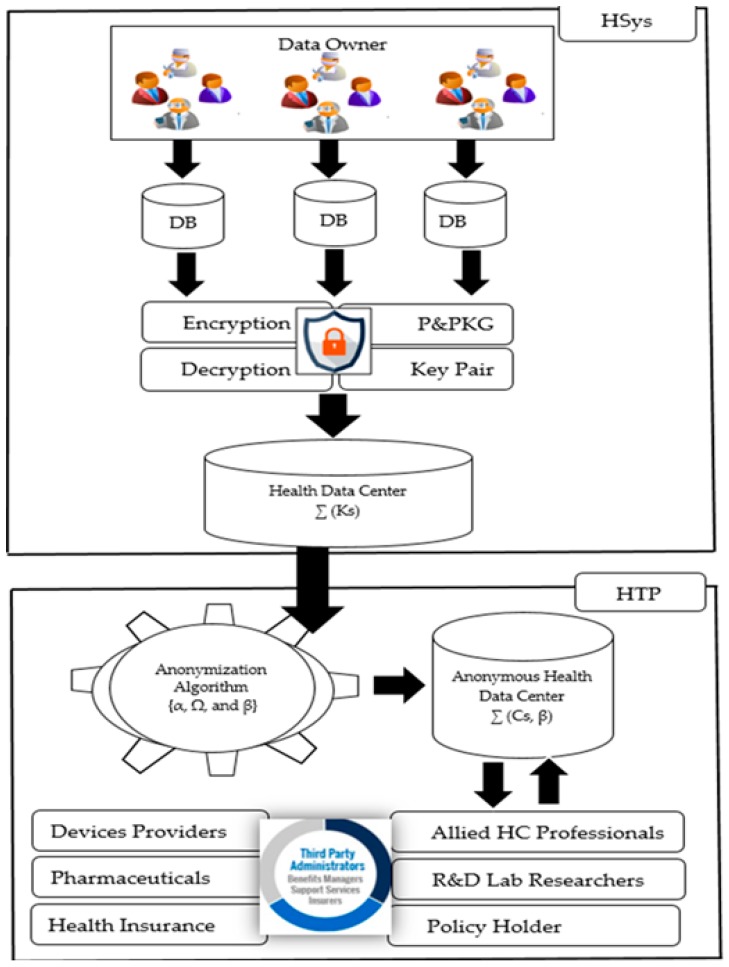
Overview of a patient’s body with IoT Medical Sensors.

**Figure 3 sensors-19-03146-f003:**
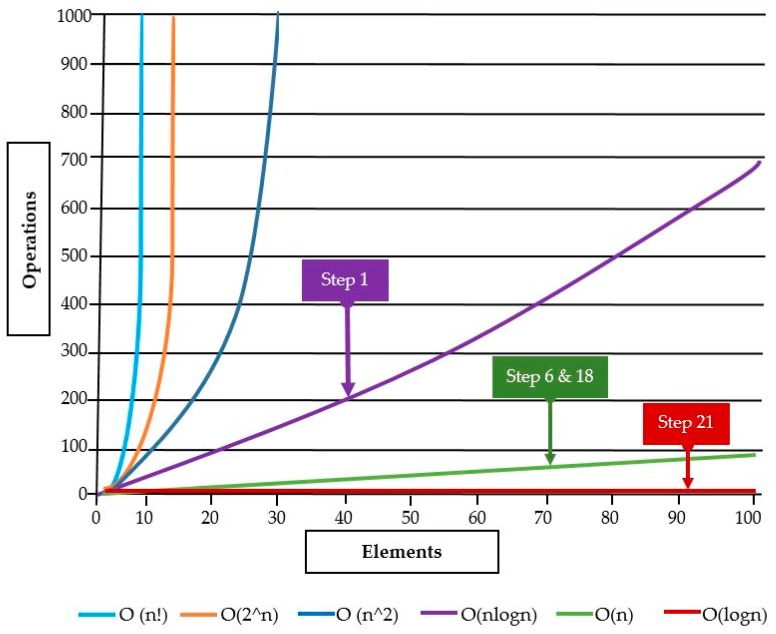
Big-O Complexity and result classification.

**Table 1 sensors-19-03146-t001:** Main parameters used in Algorithm 1.

Parameters	Description
HSys with P & PKG	Health care System with Public & Private Key Generator
Sick Person <Sp>, Physician <Ph>	Users <*U*> in the HSys
<m_s_, n_s_>	Secret Pair Key of each user
HDS	Health Data-Set
HTP	Health Third Party
∨	Or: Sp ∨ Ph

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
