# Peer review of "An IoT-Based Anonymous Function for Security and Privacy in Healthcare Sensor Networks"

_sensors, 2019, doi:10.3390/s19143146_

Reviewer 1 Report

The authors have proposed an IoT-based data anonymization framework for ensuring privacy and security in healthcare sensor networks.

The authors should consider the following points and revise the paper again.

Major issues:

(1) - Data anonymization is not a new research and there are lots of contributions in the literature, though in the paper the main focus is IoT-based connected network and streaming data. The authors should compare the proposed data anonymization approach/algorithm with existing privacy algorithms (like K-anonymity, k-clustering, differential privacy and so on), for anonymizing sensitive healthcare data.

(2) - Also, the authors did not cover access control research in this work. Using access and privacy control policies, it is possible to handle and manage sensitive medical data. The authors should review the state-of-the-art access control literature and provide a comparative analysis - including access and privacy control policies, privacy algorithms and the anonymization algorithm proposed in this work. Few of the access control papers are included here:

Recently, Context-Aware Access Control (CAAC) models have been developed, extending the basic Role-Based Access Control (RBAC) model - to manage sensitive data and determine whether users’ requests to limit data access permissions based on the contextual conditions [1-5].

Trnka and Cerny [1] have proposed a CAAC scheme based on using security levels, in which the RBAC policies are used to grant and manage data access decision...

Schefer-Wenzl and Strembeck [2] have proposed an ontology-based CAAC approach...

Hosseinzadeh et al. [3] have used OWL language and ontological techniques for modeling context-aware RBAC policies...

Colombo and Ferrari have proposed a fine-grained CAAC framework [4] - designed the access control mechanism for MongoDB, enhancing the data protection functionalities of NoSQL datastore...

Recently, Kayes et al. have introduced several CAAC models in the last few years [5-6] - that can be applicable in today's IoT-enabled smart spaces - for managing sensitive/streaming data....They have considered a wide variety of contextual conditions, for example, the situational and relationship context, utilizing the process of inferring implicit knowledge from the currently available context information.

[1] M. Trnka, and T. Cerny, On security level usage in context-aware role-based access control, Proceedings of the 31st Annual ACM Symposium on Applied Computing, pp.1192-1195, Pisa, Italy, April 04 - 08, 2016.

[2] S. Schefer-Wenzl, and M. Strembeck, Modelling context-aware RBAC models for mobile business processes, International Journal of Wireless and Mobile Computing, Vol. 6 No.5, pp. 448-462, 2013.

[3] S. Hosseinzadeh, S. Virtanen, N.D. Rodríguez, and J. Lilius, A semantic security framework and context-aware role-based access control ontology for smart spaces, Proceedings of the International Workshop on Semantic Big Data, San Francisco, California, June 26 - July 01, 2016.

[4] P. Colombo, and E. Ferrari, Enhancing NoSQL datastores with fine-grained context-aware access control: a preliminary study on MongoDB, International Journal of Cloud Computing, Vol. 6, No. 4, pp. 292-305, 753 2017.

[5] A.S.M. Kayes, J. Han, W. Rahayu, ... (2018). A Policy Model and Framework for Context-Aware Access Control to Information resources. The Computer Journal. Oxford University Press, 62(5), 670-705.

[6] A.S.M. Kayes, W. Rahayu, .... (2019). Context-Aware Access Control with Imprecise Context Characterization for Cloud-Based Data Resources. Future Generation Computer Systems. Elsevier, 93, 237-255.

(3) - Need an experimental evaluation of the proposed approach/algorithm - though, the authors have provided a mathematical analysis for accuracy proof, an experimental evaluation is really important in this work.

Minor issues:

- P1 - Please reword this - a mathematical analysis is provided as to accuracy proof...

- P2 - privacy and data protection have serious issues in term threats...need to change this sentence.

- The authors should proof-read the whole paper.

Author Response

 Dear Valued Reviewer,

Thank you for your time to review our article. Without your noble work, our article could not move forward to this stage. We really appreciate your excellent work.

Hereby, we would like to transfer our answers to your questions. Once again, with your questions, the article has been updated accordingly. Should you find any other recommendations or suggestion, please let us know, we will react promptly.

Reviewer 2 Report

  1. The Section 3 lacks the system model, the present introduction is very confused and difficult to understand.

  2. The author's main contributions should be listed in the introduction.

  3. The layout of the paper is not neat enough, especially the formula and paragraph space.

  4. Please invite someone competent in the English language and the subject matter of this manuscript go over the whole text and improve it. For example, in Section 5(line 427) , the word  the "imported" should be "important".  tIn Section 3 (line 146), the word "secret" is not in the right form.

Author Response

(The authors gave the same response as above.)

Reviewer 3 Report

Summary:
The paper "An IoT-based Anonymous Function for Security and Privacy in Healthcare Sensor Networks" proposes
an algorithm-based approach to secure healthcare-related data in an IoT setting. The authors describe, which
challenges emerge when considering healthcare data in an IoT setting. Furthermore, they discuss related work.
Then, their main two algorithms are described, including the technical aspects of the algorithms. Before concluding
their work, the authors show a mathematical proof for the correctness of their algorithms.
Points in favor:
- The paper fits to the scope of the journal
- The paper deals with a relevant topic
- The paper discusses related work
- The paper shows a mathematical proof for their algorithms
Points against the paper:
- In general, the paper is difficult to read in several aspects
  (1) Many language flaws are in the paper
      (1.1) Abstract: "In this article, ..." -> not a correct sentence
      (1.2) Abstract: "A mathematical analysis ..." -> not a correct sentence
      (1.3) Introduction : "Nowadays, ..." -> not a correct sentence
      ...
  (2) The abbreviation IBE is never explained, also PKE
      Although it s clear, it must be carefully crafted
  (3) Figure 1 must be improved and in addition, is not necessary
  (4) Section 3, with all the abbreviations, is very difficult to read, especially the algorithms
      -> a flowchart or workflow of the algorithm (its operating principle) would be more helpful
  (5) In the introduction, in the most important part, the security part, are no references given
  (6) 3.1. and 3.2., statement is not very precise
  ...
- Limitations are not discussed
- Related works should be compared in a table
- It is not provided, which aspects are the contribution, e.g., what are the new ideas of the shown algorithms
  compared to other security algorithms; on top of this, it is not clearly stated why the shown solution
  particularly fosters security in the context of healthcare data?
- Have the algorithms be applied in practice?

Author Response

Dear Valued Reviewer,

Thank you for your time to review our article. Without your noble work, our article could not move forward to this stage. We really appreciate your excellent work.

Hereby, we would like to transfer our answers to your questions. Once again, with your questions, the article has been updated accordingly. Should you find any other recommendations or suggestion, please let us know, we will react promptly.

Round  2

Reviewer 1 Report

The authors have considered all the comments and revised the paper accordingly.

Conclusion - Better to say Sensors or devices, not sensor devices.

Author Response

Dear Valued Reviewer,

Thank you for your time to review our article once again. Without your noble work, our article could not move forward to this stage. We really appreciate your excellent work.

Hereby, we would like to transfer our answers to your questions. Once again, with your questions, the article has been updated accordingly. Should you find any other recommendations or suggestion, please let us know, we will react promptly.

Reviewer 2 Report

The comments of the reviewer are well addressed. 

However,  there are still a lot of English problems in the full text.  Please make further modifications.

For example, in Section 1 (line 81), the word "as follows" while in line 90,the word is "as follow". 

So, your manuscript needs to be edited more carefully by someone with expertise in technical English editing.

 Pay particular attention to English grammar, spelling, and sentence structure so that the goals and results of the study are clear to the reader.

 In addition, the quality of figure 1 should be improved.

Author Response

(The authors gave the same response as above.)

Reviewer 3 Report

I have read the second version of the paper. I still have concerns. First, the language issues I have mentioned were not considered. Second, I cannot see where the limitations are actually discussed? Third, as mentioned, the figures must be improved in quality and size.

Finally, I would recommend to let a native speaker correct the paper.

Author Response

(The authors gave the same response as above.)
